# Dominant expansion of CD4+, CD8+ T and NK cells expressing Th1/Tc1/Type 1 cytokines in culture-positive lymph node tuberculosis

**Gokul Raj Kathamuthu**[1,2]*, **Rathinam Sridhar**[3], **Dhanaraj Baskaran**[2], **Subash Babu**[1,4]

**1** National Institutes of Health-NIRT-International Center for Excellence in Research, Chennai, India, **2** National Institute for Research in Tuberculosis (NIRT), Chennai, India, **3** Government Stanley Medical Hospital, Chennai, India, **4** Laboratory of Parasitic Diseases, National Institute of Allergy and Infectious Diseases, National Institutes of Health, Bethesda, Maryland, United States of America

* gokul.r@nirt.res.in

**Data Availability Statement:** All relevant data are within the paper and its Supporting Information files.

## Abstract

Lymph node culture-positive tuberculosis (LNTB+) is associated with increased mycobacterial antigen-induced pro-inflammatory cytokine production compared to LN culture-negative tuberculosis (LNTB-). However, the frequencies of CD4+, CD8+ T cells and NK cells expressing Th1/Tc1/Type 1 (IFNγ, TNFα, IL-2), Th17/Tc17/Type 17 (IL-17A, IL-17F, IL-22) cytokines and cytotoxic (perforin [PFN], granzyme [GZE] B, CD107a) markers in LNTB+ and LNTB- individuals are not known. Thus, we have studied the unstimulated (UNS) and mycobacterial antigen-induced frequencies of CD4+, CD8+ T and NK cells expressing Th1, Th17 cytokines and cytotoxic markers using flow cytometry. The frequencies of CD4+, CD8+ T and NK cells expressing cytokines and cytotoxic markers were not significantly different between LNTB+ and LNTB- individuals in UNS condition. In contrast, upon Mtb antigen stimulation, LNTB+ individuals are associated with significantly increased frequencies of CD4+ T cells (PPD [IFNγ, TNFα], ESAT-6 PP [IFNγ, TNFα], CFP-10 PP [IFNγ, TNFα, IL-2]), CD8+ T cells (PPD [IFNγ], ESAT-6 PP [IFNγ], CFP-10 PP [TNFα]) and NK cells (PPD [IFNγ, TNFα], ESAT-6 PP [IFNγ, TNFα], CFP-10 PP [TNFα]) expressing Th1/Tc1/Type 1, but not Th17/Tc17/Type 17 cytokines and cytotoxic markers compared to LNTB- individuals. LNTB+ individuals did not show any significant alterations in the frequencies of CD4+, CD8+ T cells and NK cells expressing cytokines and cytotoxic markers compared to LNTB- individuals upon HIV Gag PP and P/I antigen stimulation. Increased frequencies of CD4+, CD8+ T and NK cells expressing Th1/Tc1/Type 1 cytokines among the LNTB+ group indicates that the presence of mycobacteria plays a dominant role in the activation of key correlates of immune protection or induces higher immunopathology.

## Introduction

Tuberculosis (TB) is a major infectious disease with 10 million active cases and 1.3 million deaths reported worldwide [1]. Infection with *Mycobacterium tuberculosis* (Mtb) is associated with various states of disease progression such as latent TB (LTB), pulmonary TB (PTB) and

**Funding:** The author(s) received no specific funding for this work.

**Competing interests:** The authors have declared that no competing interests exist.

extra-pulmonary TB (EPTB) [2]. While the incidence of overall TB cases is falling in developing nations, the incidence of EPTB is not decreasing significantly with higher mortality and morbidity rate observed globally. EPTB represents between 15–20% of all forms of TB cases [3]. Among them, lymph node tuberculosis (LNTB) or tuberculous lymphadenitis is the typical manifestation responsible for 20% of cases with the peripheral involvement and cervical region being most usually affected [3–6]. Excisional biopsy or fine-needle aspiration cytology (FNAC) is preferred for pathologic examination; however, still, culture (either solid or liquid) or nucleic acid amplification test (NAAT) based confirmation of the *Mycobacterium* bacilli remains the gold standard for diagnosis [7]. Therefore, diagnosis of LNTB is very difficult due to the occurrence of various infectious and non-infectious diseases which are often associated with a similar clinical state. Moreover, the immune responses between the culture positive and negative LNTB phenotype remain unclear.

We have previously shown lymph node TB culture-positive (LNTB+) individuals are associated with elevated Mtb antigen-induced levels of type 1, type 17, pro-inflammatory cytokines and reduced TGFβ levels upon comparison with lymph node TB culture-negative (LNTB−) individuals [8]. Similarly, LNTB patients displayed significantly reduced CD4+, CD8+ T cells and natural killer (NK) cells expressing cytotoxic markers compared to peripheral blood [9]. We showed that LNTB individuals were associated with increased (TNFα, IL-17A) and decreased (IL-1α, IL-1β, IL-18) pro-inflammatory cytokines in lymph nodes (LN) compared to whole blood [10]. However, understanding the correlates of immune protection at the site of infection or affected LNs has not been studied. Hence, it is important to examine the relationship of diverse cytokines/cytotoxic markers and their role in immune activation at sites of Mtb infection to improve our understanding of the differences in pathogenesis between LNTB + and LNTB-. Therefore, our present study is the first to examine the cytokine and cytotoxic marker expressing CD4+, CD8+ T cells and NK cell frequencies in LNTB+ and LNTB- individuals. We demonstrate that LNTB+ individuals are predominantly associated with heightened Mtb antigen-induced CD4+, CD8+ T cells and NK cell frequencies expressing Th1/Tc1/type 1 cytokines compared to LNTB- individuals. Overall, our results indicate that the existence of mycobacteria plays a dominant role in the activation of key correlates of immune protection or induces higher immunopathology.

## Materials and methods

### Study groups

We recruited lymph node TB culture-positive (hereafter LNTB+, n = 18) and lymph node TB culture-negative (hereafter LNTB-, n = 10) patients and performed the experiments. The demographics and hematological data of the study population were reported previously [8] and the same set of lymph node (LN) samples was used for this study. LNTB+ group was characterized based on the excision biopsy exhibiting positive for *Mycobacterium tuberculosis* (Mtb) on liquid cultures and LNTB- (culture grades 0 [no or <19 colonies]) group was characterized based on the negative results for Mtb detection in liquid cultures. LNTB- individuals were diagnosed on the basis of histopathology. Both the study groups were negative for HIV infection and were not administered any steroids. The study was approved by the ethics committee of the National Institute of Research in Tuberculosis (NIRTIEC2010007). We have obtained written informed approval from both the study groups.

### Lymph node isolation, culture and antigen stimulation

The LN cells from LNTB+ and LNTB- individuals were isolated, cultured and either unstimulated, stimulated with Mtb (PPD [Staten's Serum Institute], ESAT-6, CFP10 [BEI resources,

10 µg/mL]), non-Mtb (HIV Gag PP, (AIDS Reagent Program, Division of AIDS, NIAID, NIH), 10 µg/mL) and positive control (phorbol myristate acetate/Ionomycin [P/I], (Calbiochem, San Diego, CA), 12.5 and 125 ng/mL) antigens [8, 9]. Before culturing, the LNs were transported in the Roswell Park Memorial Institute (RPMI)-1640 medium (Gibco) after biopsy and washed two times in RPMI 1640 medium. Further, the LNs were chopped into smaller pieces, treated with deoxyribonuclease (DNase, Sigma-Aldrich) and liberase (collagenase I and II, 0.1 mg/mL, Sigma-Aldrich) and incubated at 37°C for 20–30 minutes. The cells were filtered using an 80–100 µm filter (Becton Dickinson, BD), washed, centrifuged (2,600 rpm for 10 minutes) and mixed using complete RPMI (RPMI, fetal calf serum [FCS], HEPES buffer, antibiotic cocktail [gentamycin, penicillin, streptomycin]). The cells were counted using trypan blue (Bio-Whittaker[TM], Walkersville, MD) and evenly dispersed (2 million cells in 2mL/ well) in culture plates (12-well, Costar), stimulated with the above-mentioned condition. To that, Brefeldin A solution (10 µg/mL, BD) was added after 2 hours and incubated for 18 hours at 37°C. Once incubation was over, LN cells were washed with 1X PBS and fixed using cyto fix/perm (BD, 20 minutes in dark conditions). Finally, the cells were washed with 1X permeabilization buffer (Invitrogen) and stored at -80°C in phosphate buffer saline (PBS, Lonza)/ dimethyl sulphoxide (DMSO, Sigma-Aldrich) until further use.

## Extracellular and intracellular staining of lymph nodes

The LN cells were thawed at 37°C, washed with 1X PBS and spun at 2,600 rpm for 10 minutes. The supernatants were discarded and the LN cells were mixed with 1X permeabilization buffer. Further, the cells were stained first only with surface antibodies and incubated for 30–60 minutes, washed with 1X permeabilization buffer[TM] [BD Biosciences] and followed by staining with intracellular antibodies (cytokines and cytotoxic markers) and incubated overnight at 4°C. Once the incubation was over, the cells were washed with permeabilization buffer and spun at 2,600 rpm for 10 minutes. The supernatants were discarded and LN cells were mixed with 1X PBS and acquisition was performed. The surface markers used were CD3, CD4, and CD8 (all purchased from BD Biosciences) and CD56 (ebioscience[TM]). The intracellular markers used were Th1 [IFNγ (BD Biosciences), TNFα (BD Biosciences), IL-2 (ebiosciences)], Th17 [IL-17A (Miltenyi Biotech), IL-17F and IL-22 (R&D systems)] cytokines and cytotoxic markers [perforin (BD Pharmingen), granzyme B (Invitrogen- ebioscience[TM]), CD107a (BD Biosciences)]. The volume added and the other details of the antibodies were given in S1 Table. Flow cytometry (eight-colour, FACSCanto II, Diva software v.6) was performed for the sample acquisition (Becton Dickinson). Lymphocyte gating was determined by using forward vs side scatter and a total of 100,00 million cellular events were obtained. FlowJo[TM] (version 10) software was used to analyze the data and CD4[+], CD8[+] T cells and CD56[+] NK cell gating strategies for cytokines and cytotoxic markers were determined with the help of FMO. We represented our data as CD4[+], CD8[+] and NK cell frequencies expressing cytokine(s) and cytotoxic markers. The baseline (following media stimulation) values of each cytokine and cytotoxic markers were used to determine the frequency; whereas, the net frequencies were represented for Mtb, non-Mtb and PMA/I antigen stimulation.

## Analysis

The significant (P) values were calculated using the statistical software GraphPad PRISM (version 9.3) software (Graph-Pad Software, Inc., San Diego, CA). The geometric means (GM) were calculated by central tendency and the difference between the two groups was calculated using the nonparametric Mann-Whitney U test.

## Results

### LNTB+ individuals are associated with increased frequencies of CD4[+] Th1 cytokines

We show the CD4[+] and CD8[+] T cells and NK cell gating strategy in S1A Fig and FMO population of CD4, CD8 and CD56 markers in S1B Fig. The representative plots of cytokine and cytotoxic markers (CD4[+]/CD8[+] T cells and NK cells) of the study population are shown in S2 and S3 Figs. We measured the frequencies of CD4[+] T cells expressing Th1 (IFNγ, TNFα, IL-2), Th17 (IL-17A, IL-17F, IL-22) cytokines and cytotoxic (PFN, GZE B, CD107a) markers between LNTB+ and LNTB- individuals (Fig 1). We show upon unstimulated (UNS) condition (Fig 1A), the frequencies of CD4[+] T cells expressing Th1 and Th17 cytokines and cytotoxic markers were not significantly different between LNTB+ and LNTB- individuals. The frequencies of CD4[+] T cells expressing Th1 (IFNγ, TNFα and/or IL-2) cytokines but not Th17 cytokines and cytotoxic markers were significantly increased between LNTB+ and LNTB- individuals upon stimulation with Mtb (PPD, ESAT-6 PP, CFP-10 PP) antigens (Fig 1B–1D). As shown in Fig 1E, the frequencies of CD4[+] T cells expressing Th1 and Th17 cytokines and

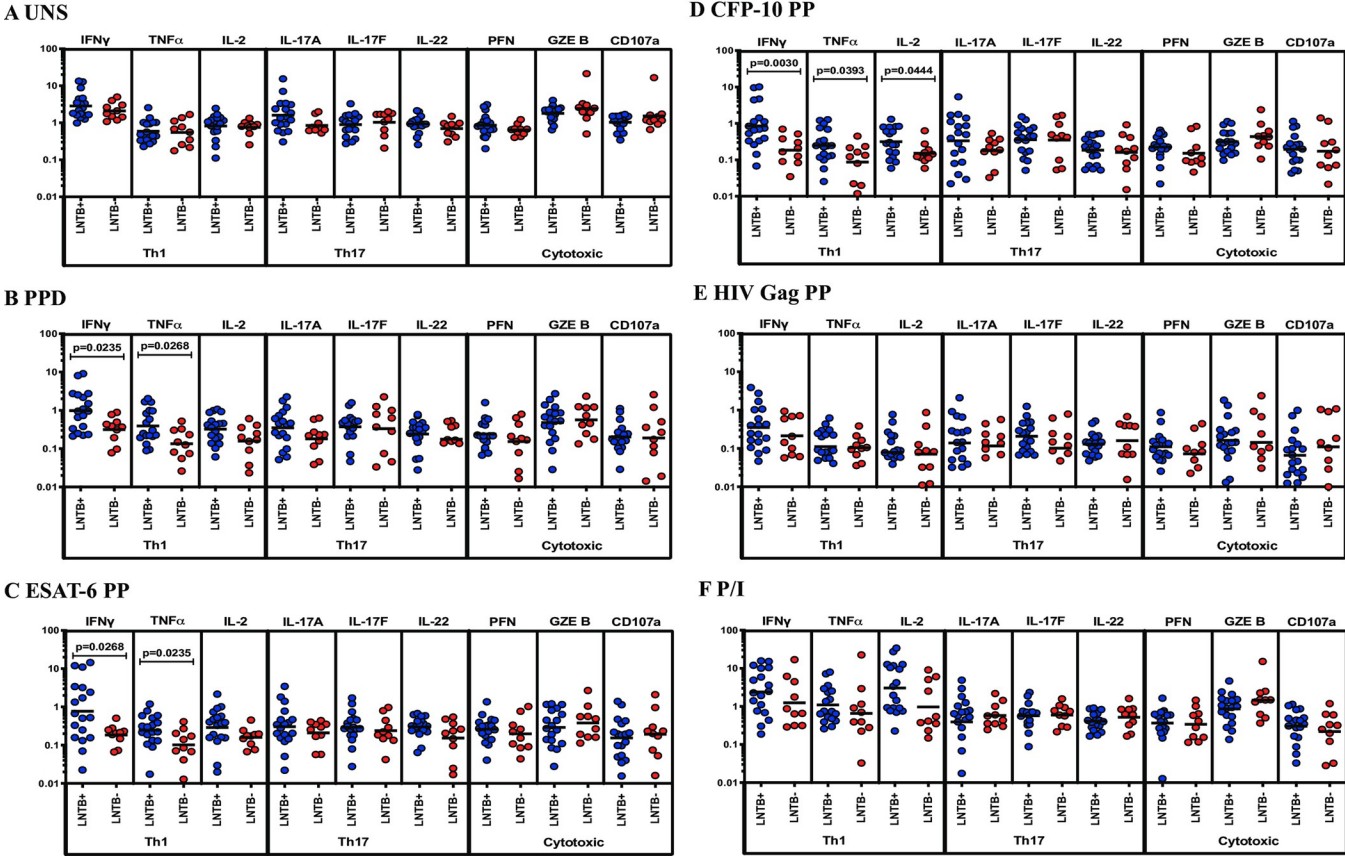

**Fig 1. LNTB+ individuals are associated with enhanced antigen-induced frequencies of CD4[+] T cells expressing Th1 (IFNγ, TNFα) cytokines.** Lymph node cells from Lymph node (LN) culture-positive tuberculosis (LNTB+, shown in blue colored round circle, n = 18) and LN culture-negative tuberculosis (LNTB-, shown in red-coloured round circle, n = 10) individuals were either unstimulated (UNS) or stimulated with mycobacterial (PPD, ESAT-6 PP, CFP-10 PP), HIV Gag PP and positive (P/I) control antigens for 18 hours. LN cells were harvested, stained and examined using multicolour flow cytometry. (A) UNS (B-D) Mtb antigen (E) HIV Gag PP and (F) positive control (P/I) antigen cultured CD4[+] T cell expressing frequencies of Th1 (IFNγ, TNFα, IL-2), Th17 (IL-17A, IL-17F, IL-22) cytokines and cytotoxic (PFN, GZE B, CD107a) markers were illustrated. The geometric mean was shown using a bar and the Mann-Whitney U test was implemented to calculate the p values and the statistically significant value is represented as p<0.05. For each individual, we have given the values as net frequencies and they were obtained by deducting the unstimulated frequencies from the antigen-stimulated frequencies.

cytotoxic markers were not significantly different between LNTB+ and LNTB- individuals upon stimulation with HIV Gag PP antigen. Finally, upon P/I antigen stimulation, CD4$^+$ T cells expressing Th1 and Th17 cytokines as well as cytotoxic marker frequencies are not significantly different between the study groups (Fig 1F). Hence, LNTB+ individuals are associated with increased frequencies of mycobacterial antigen-induced CD4$^+$ T cells expressing Th1 cytokines.

## LNTB+ individuals associated with increased frequencies of CD8$^+$ Tc1 cytokines

We analyzed CD8$^+$ T cells expressing Tc1 (IFNγ, TNFα, IL-2), Tc17 (IL-17A, IL-17F, IL-22) cytokines and cytotoxic (PFN, GZE B, CD107a) marker frequencies between LNTB+ and LNTB- individuals (Fig 2). We show the unstimulated frequencies of CD8$^+$ T cells expressing Tc1 and Tc17 cytokines and cytotoxic markers were not significantly different between LNTB + and LNTB- individuals (Fig 2A). We show the frequencies of CD8$^+$ T cells expressing Tc1 (IFNγ in PPD and ESAT-6 PP, TNFα in CFP-10 PP) cytokines but not Tc17 cytokines and cytotoxic markers were significantly increased between LNTB+ and LNTB- individuals upon

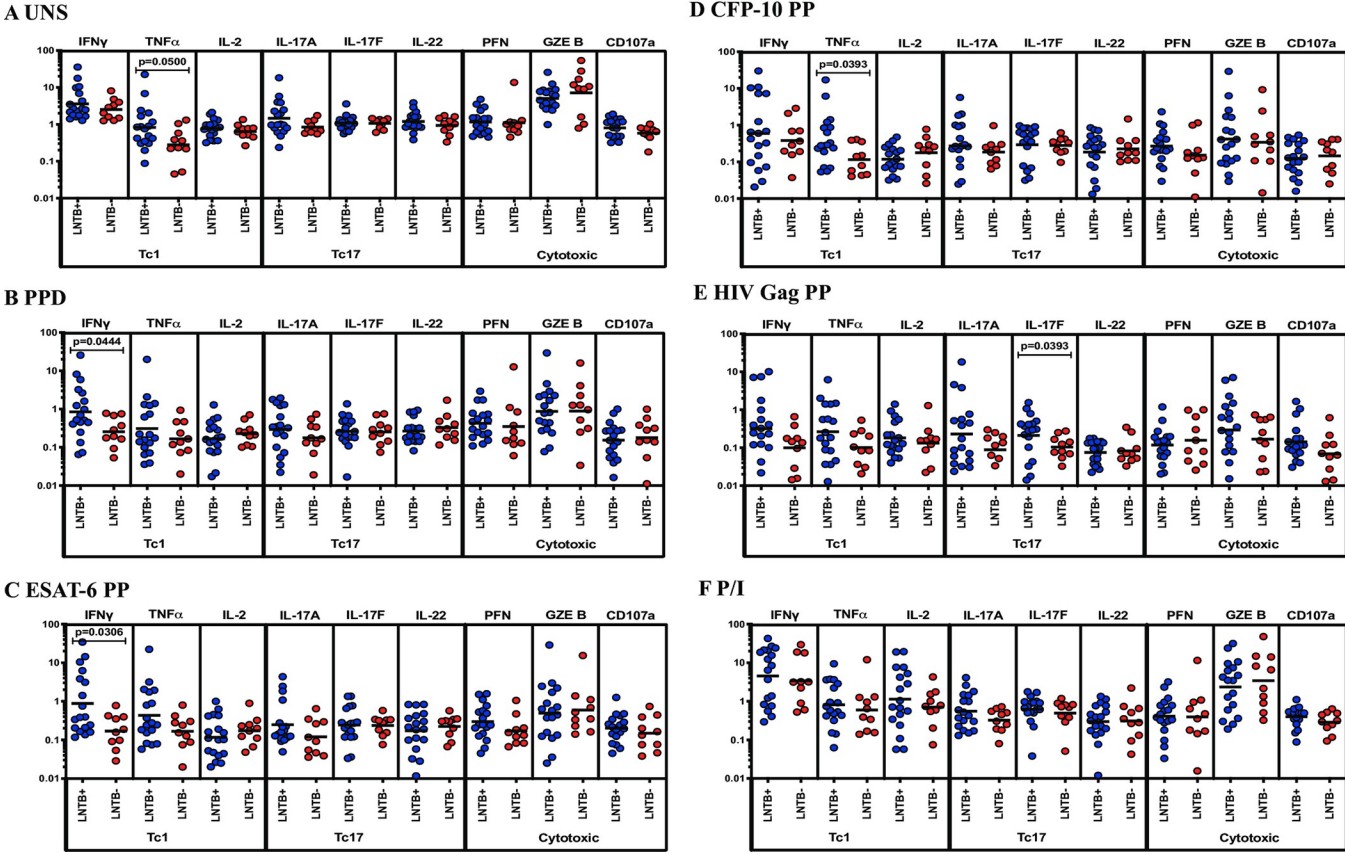

**Fig 2. LNTB+ individuals are associated with enhanced antigen-induced frequencies of CD8$^+$ T cells expressing Tc1 (IFNγ, TNFα) cytokines.** LN cells from LNTB+ (n = 18) and LNTB- (n = 10) individuals were either unstimulated (UNS) or stimulated with mycobacterial, HIV Gag PP and positive (P/I) control antigens for 18 hours. LN cells were harvested, stained and analyzed using multicolour flow cytometry. (A) UNS (B-D) Mtb antigen (E) HIV Gag PP and (F) P/I antigen-stimulated CD8$^+$ T cell expressing frequencies of Tc1 (IFNγ, TNFα, IL-2), Tc17 (IL-17A, IL-17F, IL-22) cytokines and cytotoxic (PFN, GZE B, CD107a) markers were illustrated. The geometric mean was shown using a bar and the Mann-Whitney U test was performed to calculate the p values and p<0.05 is denoted as statistically significant. For each individual, we have given the values as net frequencies and they were obtained by deducting the unstimulated frequencies from the antigen-stimulated frequencies.

Mtb antigen-induced stimulation (Fig 2B–2D). In contrast, CD8+ T cells expressing Tc1 and Tc17 (except IL-17F) cytokines and cytotoxic marker frequencies were not significantly different between LNTB+ and LNTB- individuals upon stimulation with HIV Gag PP antigen (Fig 2E). Finally, upon P/I antigen stimulation, CD8+ T cells expressing Tc1 and Tc17 cytokines and cytotoxic marker frequencies are not significantly different between the study groups (Fig 2F). Hence, LNTB+ individuals are associated with increased frequencies of mycobacterial antigen-induced CD8+ T cells expressing Tc1 cytokines.

## LNTB+ individuals are associated with increased frequencies of NK cells expressing Type 1 cytokines

We analyzed the frequencies of NK cells expressing Type 1 (IFNγ, TNFα, IL-2), type 17 (IL-17A, IL-17F, IL-22) cytokines and cytotoxic (PFN, GZE B, CD107a) markers between LNTB + and LNTB- individuals (Fig 3). We show the unstimulated frequencies of NK cells expressing Type 1 and Type 17 cytokines and cytotoxic markers were not significantly different between LNTB+ and LNTB- individuals (Fig 3A). As we have shown in Fig 3B–3D, the frequencies of NK cells expressing Type 1 (IFNγ and TNFα in PPD and ESAT-6 PP, IFNγ in CFP-10 PP)

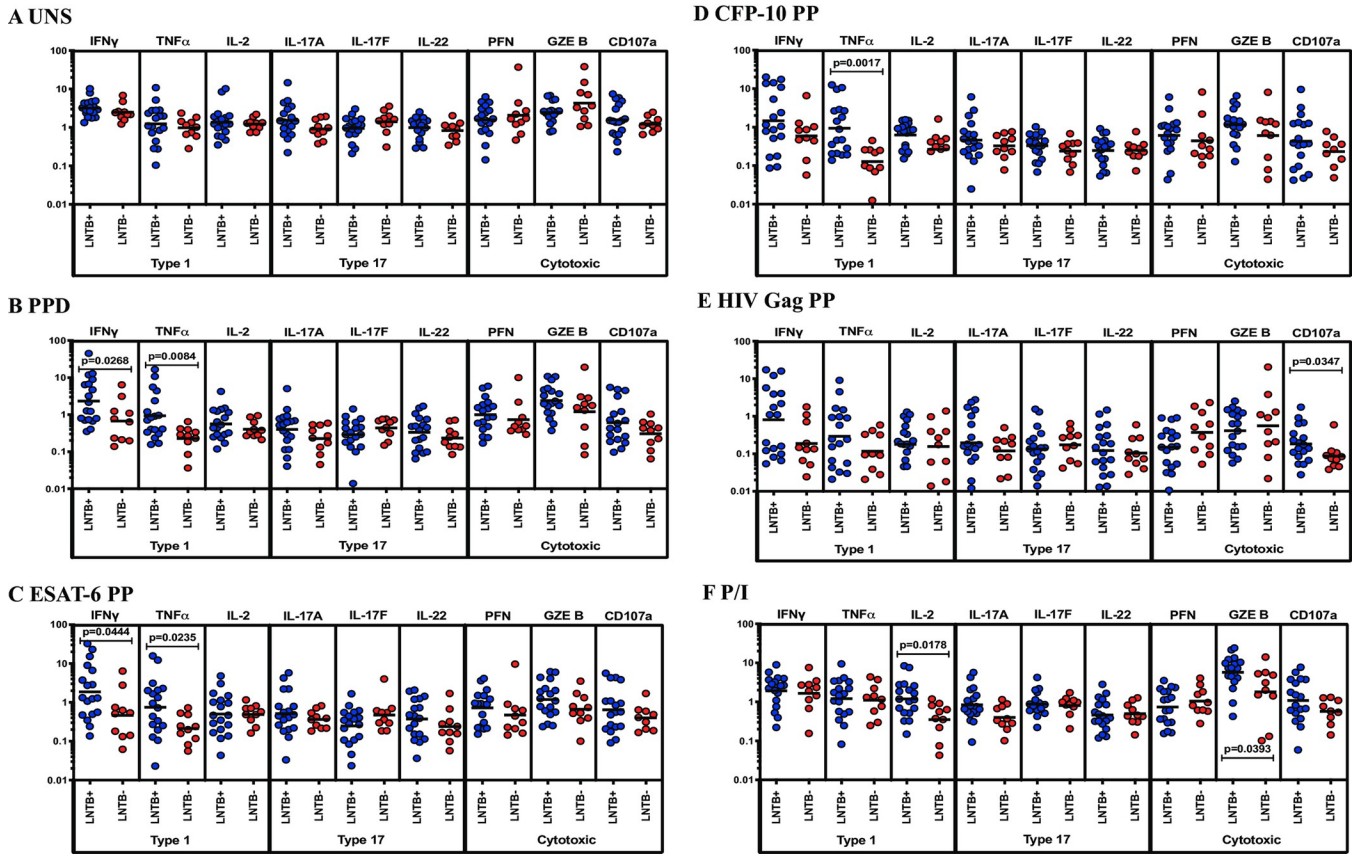

**Fig 3. LNTB+ individuals are associated with enhanced antigen-induced frequencies of NK cells expressing type 1 (IFNγ, TNFα) cytokines.** LN cells from LNTB+ (n = 18) and LNTB- (n = 10) individuals were either unstimulated (UNS) or stimulated with mycobacterial (PPD, ESAT-6 PP, CFP-10 PP), HIV Gag PP and positive (P/I) control antigens for 18 hours. LN cells were harvested, stained and analyzed using multicolour flow cytometry. (A) UNS (B-D) Mtb antigen (E) HIV Gag PP and (F) P/I antigen-stimulated CD4+ T cell expressing frequencies of type 1 (IFNγ, TNFα, IL-2), type 17 (IL-17A, IL-17F, IL-22) cytokines and cytotoxic (PFN, GZE B, CD107a) markers were illustrated. Geometric mean was shown using a bar and Mann-Whitney U test was performed to calculate the p values and p<0.05 is denoted as statistically significant. For each individual, we have given the values as net frequencies and they were obtained by deducting the unstimulated frequencies from the antigen-stimulated frequencies.

cytokines but not Type 17 cytokines and cytotoxic markers were significantly increased in LNTB+ compared to LNTB- individuals upon Mtb antigen-induced stimulation. In contrast, NK cells expressing Type 1 and Type 17 cytokines and cytotoxic (except CD107a) marker frequencies were not significantly different between LNTB+ and LNTB- individuals upon stimulation with HIV Gag PP antigen (Fig 3E). Finally, upon P/I antigen stimulation, NK cells expressing Type 1 and Type 17 cytokines and cytotoxic frequencies are not significantly different between the study groups (Fig 3F). Hence, LNTB+ individuals are associated with increased Mtb antigen-induced NK cells expressing Type 1 cytokine frequencies.

## Discussion

Investigating the immune-mediated cytokines and cytotoxic markers produced at the infection site or affected LNs is crucial and provides knowledge on whether they can mediate immune protection or induce pathology. Previous studies have shown that LNs are the collective place of Mtb infection in both cattle and non-human primates [11, 12]. We and others have shown the immunological responses associated with LNs of Mtb infected patients and compared them with biopsied cervical LNTB, healthy individuals, pulmonary TB and other diseases [8, 13–15]. However, the knowledge of pathogen-specific tissue-related immune alterations induced by bacteriological burden is scarce. Also, there are no reports have shown the Mtb antigen-induced frequencies of CD4+, CD8+ T cells and NK cells expressing Th1/Tc1/type 1, Th17/Tc1/type 17 cytokines and cytotoxic markers among culture-positive (LNTB+) and culture-negative (LNTB-) LNTB disease. Thus, we have studied the same in our present study and shown that LNTB+ individuals are associated with enhanced Th1/Tc1/type 1 (IFNγ, TNFα) cytokines (not the other cytokines and cytotoxic markers) compared to LNTB- individuals.

CD4+ T cells produce both Th1 and Th17 cytokines to control the disease [16]. Th1 immunity, especially both IFNγ and TNFα plays an instrumental role in the progress of defensive immune responses in the fight against TB disease [17, 18]. Th17 cells function by the induction of IL-17A and IL-17F cytokines which are instrumental in facilitating cellular immunity to both extra and intracellular microbes, including Mtb [19, 20]. CD4+ T cells induce cytotoxic mechanisms by secreting cytotoxic (granzyme B and perforin) granules and might kill the target cells in an antigen-specific fashion through direct interaction and able to recognize some explicit marker proteins or transcription factors [21]. Our data exhibit significantly elevated frequencies of Th1 (IFNγ and TNFα) cytokines in LNTB+ individuals compared to LNTB- individuals. The frequencies are mycobacterial antigen-specific; since there are no significant differences were observed either in the unstimulated condition or in the HIV antigen stimulation. Our earlier data have also shown similar findings on type 1 cytokines in LNTB+ than LNTB− group upon Mtb antigen stimulation [8]. Our data also suggests that there are no significant differences were observed in the IL-2 or Th17 cytokines and cytotoxic markers in both unstimulated and antigen-stimulated conditions. Overall, we show the crucial role of Th1 expressing cytokines in the disease pathogenesis of LNTB. The higher frequencies of Th1 cytokines could enhance disease severity or might reflect the increased antigen or bacterial load in the LNs of LNTB [8, 10].

Similar to CD4+ Th1 cells, CD8+ Tc1 cells express Tc1 (IFNγ, TNFα, IL-2) cytokines which are essential during Mtb infection [22]. Along with the above cytokines CD8+ T cells might induce Tc17 cytokines, especially IL-17 which have been shown in TB disease [23]. Notably, CD8+ T cells can induce cytolytic ability through granule (perforin, granzymes, granulysin) production to kill Mtb directly or Fas-Fas ligand interaction to stimulate apoptosis [22]. We previously described that the LNTB+ group is known to associate with increased Mtb antigen-

stimulated type 17 (IL-17F, IL-22) cytokines than LNTB– group [8]. We show that the frequencies of CD8+ T cell expressing Tc1 (IFNγ, TNFα) cytokines were significantly elevated in LNTB+ individuals compared to LNTB- individuals. Again, our results illustrate higher frequencies of CD8+ T cell expressing Tc1 cytokines potentially indicating greater Mtb antigen load and bacterial burden in LNTB infection [24].

NK cell-mediated responses are crucial during the initial phase of immune protection against intracellular microbes. They also play an important function in linking innate and adaptive immune responses and could produce type 1 and type 17 cytokines [25, 26]. Our study has shown that NK cells expressing type 1 cytokine frequencies were significantly enhanced in LNTB+ individuals compared to LNTB- individuals. The changes observed in the frequencies of type 1 cytokines are pathogen-specific. We previously revealed that cytotoxic markers were significantly decreased in LNs compared to peripheral blood in LNTB disease [9]. Reduced frequencies of type 1 cytokines might be associated with poor clearance of mycobacteria and impaired activation of innate immunity therefore associated with enhanced disease pathogenesis in LNTB+ individuals compared to LNTB- individuals [8, 27].

Overall, we show that culture-positive LNs have increased CD4/CD8/NK cells expressing Th1/Tc1/type 1 cytokine responses that might occur due to increased bacteriological burden in the culture-positive affected LNs. However, our study has certain limitations by not examining the LNs of the normal healthy individuals or the other chronic disease patients which are of future interest. The other future interest is to examine the Th2 cytokine frequencies between the LNTB+ and LNTB- population. Also, the other limitation of our study is being cross-sectional, not a longitudinal study, and showing the data with a smaller sample size.

## Supporting information

**S1 Fig. Lymph node tuberculosis (LNTB) gating strategy for CD4+, CD8+ T cells and NK cells.** (A) Lymphocytes are gated on LN sample and further gated on single cells. From single cells CD3+ T cells and NK (CD3-CD56+) cell population were gated. CD3+ T cells were further gated for CD4+ and CD8+ T cells. (B) FMO population of CD4+, CD8+ T cells and NK cells. (PDF)

**S2 Fig. LNTB representative plots of CD4+/CD8+ T cells.** (A) Th1 (IFNγ, TNFα, IL-2) cytokines, (B) Th17 (IL-17A, IL-17F, IL-22) cytokines, (C) cytotoxic (PFN, GZE B, CD107a) markers upon UNS, Mtb (PPD) antigen stimulation and positive antigen (P/I) control stimulation. (PDF)

**S3 Fig. LNTB representative plots of NK cells.** (A) Th1 (IFNγ, TNFα, IL-2) cytokines, (B) Th17 (IL-17A, IL-17F, IL-22) cytokines, (C) cytotoxic (PFN, GZE B, CD107a) markers upon UNS, Mtb (PPD, CFP-10 PP) antigen stimulation and positive antigen (P/I) control stimulation. (PDF)

**S1 Table. Antibodies panel used in the study.** (DOC)

## Acknowledgments

The authors thank V. Rajesh Kumar of NIH-ICER-NIRT, the staff of the Department of Clinical Research, NIRT, Government Stanley Hospital, Government General Hospital and Government Kilpauk Medical Hospital, Chennai for valuable assistance in recruiting the patients

for this study. This research work was supported by the Division of Intramural Research, National Institute of Allergy and Infectious Diseases (NIAID).

## Author Contributions

**Conceptualization:** Gokul Raj Kathamuthu, Subash Babu.

**Data curation:** Gokul Raj Kathamuthu.

**Formal analysis:** Gokul Raj Kathamuthu.

**Funding acquisition:** Subash Babu.

**Investigation:** Gokul Raj Kathamuthu, Subash Babu.

**Methodology:** Subash Babu.

**Project administration:** Gokul Raj Kathamuthu, Subash Babu.

**Resources:** Rathinam Sridhar, Dhanaraj Baskaran, Subash Babu.

**Software:** Subash Babu.

**Supervision:** Subash Babu.

**Validation:** Subash Babu.

**Writing – original draft:** Gokul Raj Kathamuthu.

**Writing – review & editing:** Gokul Raj Kathamuthu, Subash Babu.

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
