## [Decision Letter · Decision Letter 0]

3 Feb 2022

PONE-D-21-36622Dominant expansion of CD4+, CD8+ T and NK cells expressing Th1/Tc1/Type 1 cytokines in culture-positive lymph node tuberculosisPLOS ONE

Dear Dr. Kathamuthu,

Thank you for submitting your manuscript to PLOS ONE. After careful consideration, we feel that it has merit but does not fully meet PLOS ONE’s publication criteria as it currently stands. Therefore, we invite you to submit a revised version of the manuscript that addresses the points raised during the review process.

We look forward to receiving your revised manuscript.

Kind regards,

Angelo A. Izzo

Academic Editor

PLOS ONE

https://journals.plos.org/plosone/s/file?id=ba62/PLOSOne_formatting_sample_title_authors_affiliations.pdf"

Reviewers' comments:

Reviewer's Responses to Questions

**Comments to the Author**

1. Is the manuscript technically sound, and do the data support the conclusions?

Reviewer #1: Partly

Reviewer #2: Partly

2. Has the statistical analysis been performed appropriately and rigorously? 

Reviewer #1: Yes

Reviewer #2: No

3. Have the authors made all data underlying the findings in their manuscript fully available?

Reviewer #1: Yes

Reviewer #2: Yes

4. Is the manuscript presented in an intelligible fashion and written in standard English?

Reviewer #1: No

Reviewer #2: Yes

5. Review Comments to the Author

Reviewer #1: In general, the manuscript needs significant review before submission. Authors should proof read and correct all grammar before any submission. There is also lack of consistency in the abbreviations used throughout the manuscript. Abbreviations used to indicate LNTB+ and – should be consistent. Methods section lacks details of protocols and reagents used. All the figures need to follow consistent formatting. Unfortunately, the manuscript is not up to standard of publishing yet.

Reviewer #2: Comments

The introduction section should state clearly what is the hypothesis and justification of the study. It is not clear why the comparisons made by the authors between LN+ and LN- extra pulmonary tuberculosis contribute to a better understanding of anti-TB immunity. This is relevant since one of the caveats of the study is the lack of analysis of specimens from healthy individuals.

Lines 102-107. Please add the rationale for this conclusion.

Lines 103-104. I won’t call these cells antigen specific cells just because they expressed cytokines after stimulation with Mtb components. To do so, the authors should have carried out tetramer staining.

Methods are short. Despite authors refer to a previous investigation, they should have at least described the origin and disease characteristics of the individuals who donated lymph node tissues. Also, did you stain lymph node samples for Mtb to be completely sure that LN- do not have bacilli?

Although the limitation is acknowledged, the authors should try to analyze healthy control samples.

Results:

Supplementary figure 1. The gate on NK cells is missing CD56 high NK cells. Also, it is curious that you show a big and discrete population of CD3+CD56+ cells. Can you add a plot showing the FMO controls for this analysis?

Supplementary figures 2 and 3 are not necessary but the authors should definitively illustrate the differences they claimed by adding the correspondent plots to figures 1-3

Did you analyze the frequency of T cells expressing both Th1 and Th2 cytokines?

Discussion should clarify how these data are relevant for the study of extra pulmonary tuberculosis and what is the gap in knowledge covered by this investigation. What’s different from previous analyses of lymph nodes with TB? What does it mean to find increased cytokine and cytotoxic functions of T and NK cells only in samples with viable Mtb but not Mtb- samples?

6. PLOS authors have the option to publish the peer review history of their article (what does this mean?). If published, this will include your full peer review and any attached files.

Reviewer #1: No

Reviewer #2: **Yes: **Dr. Joaquin Zúñiga

---

## [Author Response · Author response to Decision Letter 0]

15 Feb 2022

Dear Reviewers,

Thanks for your valuable comments for our manuscript. We have given the responses to all the comments raised by both the reviewers.

Comments Response Line numbers revised

Reviewer 1 

In general, the manuscript needs significant review before submission. Authors should proof read and correct all grammar before any submission. There is also lack of consistency in the abbreviations used throughout the manuscript. Abbreviations used to indicate LNTB+ and – should be consistent. Methods section lacks details of protocols and reagents used. All the figures need to follow consistent formatting. Unfortunately, the manuscript is not up to standard of publishing. 

 As suggested by the reviewer now we have done grammar corrections throughout the manuscript and uniform abbreviations are incorporated and figures are formatted. Multiple places

Abstract: 

Abstract should not be in split into sections. Please combine background, methods, results and conclusions together and submit as one paragraph. As suggested by the reviewer we have now combined the abstract section and given as a single paragraph. Lines 27-47

Line 30: It should be (LNTB-) We apologize and now we have corrected LNTB- in the line number 30. Now it is in Line 29.

Introduction: 

Introduce what LN stands for e.g. lymph node (LN) before using abbreviation. If you’re using LN for lymph node throughout manuscript, please use this abbreviation. As suggested by the reviewer we have already included the expansion of lymph node tuberculosis (LNTB) in the introduction. line number 70.

Briefly explain the significance of your research and why it’s important to distinguish between LNTB+ and – As suggested by the reviewer now have given our study significance in the introduction as “Hence, it is important to examine the relationship of diverse cytokines/cytotoxic markers and their role in immune activation at sites of Mtb infection to improve our understanding of the differences in pathogenesis between LNTB+ and LNTB-“. Line 91, 151-154

Line 95-97, TBL LNs are you referring to LNTB+? Avoid using TBL and LNTB+ interchangeably. Stick to one term As suggested by the reviewer now we have removed TBL and used LNTB here and thereafter throughout the manuscript. Now it is in line 87 and other places

Line 98- provide e.g. of increased cytokines and decreased cytokines. Otherwise your statement is too vague. As suggested by the reviewer now we have revised the sentence as “We showed that LNTB individuals were associated with increased (TNFa, IL-17A) and decreased (IL-1a, IL-1b, IL-18) pro-inflammatory cytokines than whole blood”. Now it is in line 90

Methods:

Lymph node isolation, culture and antigen stimulation. 

How were the lymph nodes extracted? Were cells purified before use? If so what component? Or were they used directly?

 As suggested by the reviewer now we have included the extraction procedure. The cells were filtered using 80-100 μm filter (Becton Dickinson, BD). It is now included in the materials and methods section.

Where the lymph nodes used for culture immediately after removal or were they stored and used later? Please indicate these steps No, the cells were stored in -80o C in 1X phosphate buffer saline (PBS)/ dimethyl sulphoxide (DMSO). Then the cells were thawed at 37o C and washed with PBS before staining. It is now included in the materials and methods section.

Authors need to mention concentrations of each stimulation used. As suggested by the reviewer now we have included the concentrations It is now included in the materials and methods section.

Please detail all exact protocols used. As suggested by the reviewer now we have given the complete protocol. It is now included in the materials and methods section.

Make heading of this section “Extra cellular and intra cellular staining of lymph nodes”.

 As suggested by the reviewer now we have changed the title as “Extra cellular and intra cellular staining of lymph nodes”. It is now included in the materials and methods

Please provide list of antibodies used in the form of a supplementary table, with cat#, manufacturer, volume of antibody used in final volume. 

 We have provided the list of antibodies, catalogue number, manufacturer details and antibody volume in Supplementary table Supplementary table 1

Provide detail protocols, i.e. step by step on the antibody staining procedure. 

 As suggested by the reviewer now we have given the detailed staining procedure. It is now included in the materials and methods

Mention the manufacturer names of the permeabilization buffer and media used. 

 As suggested by the reviewer now we have given the manufacturer names of the permeabilization buffer (Invitrogen) and RPMI media (Gibco). It is now included in the materials and methods

Provide recipes of cell culture media and any and all buffers with their respective manufacturer names.

 As suggested by the reviewer now we have included the manufacturer names of cell culture media and buffers It is now included in the materials and methods

 What is the FlowJo version number?

 As suggested by the reviewer now we have included the FlowJo version number It is now included in the materials and methods

Results: 

Figure 1: Use different colours to indicate two cohorts (black/red). Axis labels are too small and difficult to read. Please increase text size. Use consistent labeling to indicate LNTB+/- not cult+ and cult – on figures. This is applicable to entire manuscript where authors are not consistent with using the abbreviations. 

The same comments apply to figure 2 and 3. 

 As suggested by the reviewer now we have revised the figures and also changed the x-axis and throughout the manuscript from cul+/- to LNTB+/-. 

 Figures 1, 2, 3

Supplementary figures 1, 2, 3: Use same format for gating strategy. If using pseudo colour, stick to pseudo colour if not stick to contour plots for all three. Increase font size of the labels and axis. Clearly mark in each plot the population of cells you’re referring to and use consistent formatting. Use the same formatting for every figure. Keep it consistent. 

 As suggested by the reviewer, now we have given the supplementary figures (1, 2, 3) in pseudo colour and increased the font size of the labels.

 Supplementary Figures 

Figure 2 and 3: Show one representative plot from an unstimulated for negative and one positive control for each cytokine to give readers an idea of where the gates were set. You do not need to show plots for each stim condition. 

 As suggested by the reviewer, now we have revised the representative plots and now included one negative (UNS), one antigen stimulation (PPD) and one positive control stimulation (P/I) plots and shown in the supplementary figure 1A.

 Supplementary Figures 2, 3

Discussion: 

Line 303-305: How did you arrive to this conclusion? Where are the ref papers to back up your claim?

 As suggested by the reviewer now we have included the reference number in the line number (now the line number is 403-404). 

 Discussion section

Line 314-316: Ref?

 As suggested by the reviewer now we have included the reference number in the line number (now the line number is 413-415). 

 Discussion section

Line 324-326: ref? As suggested by the reviewer now we have included the reference number in the line number (now the line number is 418, 437-439). 

 Discussion section

Lines 102-107. Please add the rationale for this conclusion.

 As suggested by the reviewer, now we have included the rationale as: However, understanding the correlates of immune protection at the site of infection or affected LNs has not been studied. Hence, it is important to examine the relationship of diverse cytokines/cytotoxic markers and their role in immune activation at sites of Mtb infection to improve our understanding of the differences in pathogenesis between LNTB+ and LNTB-. Introduction section

Reviewer 2

The introduction section should state clearly what is the hypothesis and justification of the study. It is not clear why the comparisons made by the authors between LN+ and LN- extra pulmonary tuberculosis contribute to a better understanding of anti-TB immunity. This is relevant since one of the caveats of the study is the lack of analysis of specimens from healthy individuals

Comments Response Line numbers revised

Lines 102-107. Please add the rationale for this conclusion.

 As suggested by the reviewer, now we have included the rationale as “Hence, it is important to examine the relationship of diverse cytokines/cytotoxic markers and their role in immune activation at sites of Mtb infection to improve our understanding of the differences in pathogenesis between LNTB+ and LNTB”. Introduction section

Lines 103-104. I won’t call these cells antigen specific cells just because they expressed cytokines after stimulation with Mtb components. To do so, the authors should have carried out tetramer staining. As suggested by the reviewer, now we have changed the antigen specific to antigen induced in the introduction section.

 Introduction section and multiple places in the manuscript

Methods are short. Despite authors refer to a previous investigation, they should have at least described the origin and disease characteristics of the individuals who donated lymph node tissues. Also, did you stain lymph node samples for Mtb to be completely sure that LN- do not have bacilli? As suggested by the reviewer now we have given the detailed staining procedure. Yes, we did the staining for both the groups (LNTB+, LNTB-). LNTB+ group were characterized based on the excision biopsy exhibiting positive for Mycobacterium tuberculosis (Mtb) on liquid cultures and LNTB- (culture grades 0 [no or <19 colonies]) group were characterized based on the negative results for Mtb detection in liquid cultures. LNTB- individuals were diagnosed on the basis of histopathology. Materials and methods section

Although the limitation is acknowledged, the authors should try to analyze healthy control samples.

 Yes, we agree with the reviewer point, but there is no feasibility or ethical committee approval to obtain the lymph nodes from healthy individuals. Thus, we have included as a limitation. Discussion section

Results:

Supplementary figure 1. The gate on NK cells is missing CD56 high NK cells. Also, it is curious that you show a big and discrete population of CD3+CD56+ cells. Can you add a plot showing the FMO controls for this analysis?

 As suggested by the reviewer, now we have gated the CD56 high population and included in the supplementary figure 1A and we have included the FMO of CD4, CD8 and CD56 population in the supplementary figure 1B. 

 Supplementary figure A and 1B

Supplementary figures 2 and 3 are not necessary but the authors should definitively illustrate the differences they claimed by adding the correspondent plots to figures 1-3

 The other reviewer suggested to include one from negative, Mtb antigen and positive antigen stimulation. Thus, now we included UNS, PPD and P/I plot alone in the Supplementary figures 2 and 3.

 Supplementary figures

Did you analyze the frequency of T cells expressing both Th1 and Th2 cytokines?

 No, we have measured the frequencies of Th1 cytokines alone and we will try to measure the Th2 cytokines in our future studies. It is now included in the discussion section as “The other future interest is to examine the Th2 cytokine frequencies between the LNTB+ and LNTB- population”. Discussion section

Discussion should clarify how these data are relevant for the study of extra pulmonary tuberculosis and what is the gap in knowledge covered by this investigation. What’s different from previous analyses of lymph nodes with TB? 

 We now given the relevance of the study as “However, the study of pathogen-specific tissue related immune alterations induced by bacteriological burden is scarce”. It is now included in the discussion section. Discussion section

What does it mean to find increased cytokine and cytotoxic functions of T and NK cells only in samples with viable Mtb but not Mtb- samples?

 It is not the Mtb- samples, we compared the cytokines and cytotoxic markers between lymph node (LNs) and whole blood of same individuals infected with Mtb and shown there was significant alterations were observed between the LNs and whole blood [8, 9, in the introduction]. Introduction references 8, 9.

---

## [Decision Letter · Decision Letter 1]

8 Mar 2022

PONE-D-21-36622R1Dominant expansion of CD4+, CD8+ T and NK cells expressing Th1/Tc1/Type 1 cytokines in culture-positive lymph node tuberculosisPLOS ONE

Dear Dr. Kathamuthu,

Thank you for submitting your manuscript to PLOS ONE. After careful consideration, we feel that it has merit but does not fully meet PLOS ONE’s publication criteria as it currently stands. Therefore, we invite you to submit a revised version of the manuscript that addresses the points raised during the review process.

Please pay careful attention to comments made by Reviewer #1.

We look forward to receiving your revised manuscript.

Kind regards,

Angelo A. Izzo

Academic Editor

PLOS ONE

Journal Requirements:

Additional Editor Comments (if provided):

Reviewers' comments:

Reviewer's Responses to Questions

**Comments to the Author**

1. If the authors have adequately addressed your comments raised in a previous round of review and you feel that this manuscript is now acceptable for publication, you may indicate that here to bypass the “Comments to the Author” section, enter your conflict of interest statement in the “Confidential to Editor” section, and submit your "Accept" recommendation.

Reviewer #1: All comments have been addressed

Reviewer #2: All comments have been addressed

2. Is the manuscript technically sound, and do the data support the conclusions?

Reviewer #1: Yes

Reviewer #2: Yes

3. Has the statistical analysis been performed appropriately and rigorously? 

Reviewer #1: Yes

Reviewer #2: Yes

4. Have the authors made all data underlying the findings in their manuscript fully available?

Reviewer #1: Yes

Reviewer #2: Yes

5. Is the manuscript presented in an intelligible fashion and written in standard English?

Reviewer #1: Yes

Reviewer #2: Yes

6. Review Comments to the Author

Reviewer #1: Overall there is a definite improvement in the new draft. However, authors will benefit from another round of proof reading before the final submission.

211 T cells expressing Tc1 and Tc17 (expect IL-17F) cytokines and cytotoxic marker frequencies: change expect to except

213 Finally, upon P/I antigen stimulation, CD4+ 213 T cells 214 expressing Tc1 and Tc17 cytokines and cytotoxic marker frequencies are not significantly different between the study groups (Figure 2F).

This is CD8 and not CD4.

Figure 1-3 would benefit from a legend to state which stimulation condition is demonstrated in each row of graphs. This will help reader easily navigate and identify plots.

Discussion

As suggested by the reviewer now we have included the reference number in the line number (now the line number is 413-415). Discussion section Line 324-326: ref?As suggested by the reviewer now we have included the reference number in the line number (now the line number is 418, 437-439). Discussion section

Line 413-415, 418, 437-439 does not indicate to the correct lines in this version of the draft. Please indicate the correct line numbers for reviewers in the future.

Reviewer #2: Thank you to the authors because they addressed all concerns and the quality of the manuscript was improved.

7. PLOS authors have the option to publish the peer review history of their article (what does this mean?). If published, this will include your full peer review and any attached files.

Reviewer #1: No

Reviewer #2: **Yes: **Dr. Joaquin Zúñiga

---

## [Author Response · Author response to Decision Letter 1]

10 Mar 2022

Reply to Reviewer,

Reviewer #1: Overall there is a definite improvement in the new draft. However, authors will benefit from another round of proof reading before the final submission.

Response: Thank you for the valuable comments on our manuscript.

211 T cells expressing Tc1 and Tc17 (expect IL-17F) cytokines and cytotoxic marker frequencies: change expect to except

Response: We apologize for the spelling error and now we changed expect to except.

213 Finally, upon P/I antigen stimulation, CD4+ 213 T cells 214 expressing Tc1 and Tc17 cytokines and cytotoxic marker frequencies are not significantly different between the study groups (Figure 2F).

This is CD8 and not CD4.

Response: We apologize for the typographical mistake and now we changed CD4 to CD8.

Figure 1-3 would benefit from a legend to state which stimulation condition is demonstrated in each row of graphs. This will help reader easily navigate and identify plots.

Response: As suggested by the reviewer now we included the legends for the stimulation in figures 1-3.

Discussion

As suggested by the reviewer now we have included the reference number in the line number (now the line number is 413-415). Discussion section Line 324-326: ref? 

Response: We have included the reference (reference number 24) in line number 324-326. 

24. Kumar NP, Sridhar R, Hanna LE, Banurekha VV, Jawahar MS, Nutman TB, Babu S. Altered CD8(+) T cell frequency and function in tuberculous lymphadenitis. Tuberculosis (Edinb). 2014; 94(5):482-93. doi: 10.1016/j.tube.2014.06.007. 

As suggested by the reviewer now we have included the reference number in the line number (now the line number is 418, 437-439). Discussion section

Line 413-415, 418, 437-439 does not indicate to the correct lines in this version of the draft. Please indicate the correct line numbers for reviewers in the future.

Response: We apologize and now we have included the correct line numbers (line numbers 292-294 and 302-304 in the clean version of the manuscript) along with the references.

24. Kumar NP, Sridhar R, Hanna LE, Banurekha VV, Jawahar MS, Nutman TB, Babu S. Altered CD8(+) T cell frequency and function in tuberculous lymphadenitis. Tuberculosis (Edinb). 2014; 94(5):482-93. doi: 10.1016/j.tube.2014.06.007. 

8. Kathamuthu GR, Moideen K, Sridhar R, Baskaran D, Babu S. Enhanced mycobacterial antigen-induced pro-inflammatory cytokine production in lymph node tuberculosis. Am J Trop Med Hyg. 2019; 100(6):1401-1406. doi:10.4269/ajtmh.18-0834

27. Kathamuthu GR, Kumar NP, Moideen K, Sridhar R, Baskaran D, Babu S. Diminished type 1 and type 17 cytokine expressing - Natural killer cell frequencies in tuberculous lymphadenitis. Tuberculosis (Edinb). 2019; 118:101856. doi: 10.1016/j.tube.2019.101856.

---

## [Editor Report · Decision Letter 2]

16 May 2022

Dominant expansion of CD4+, CD8+ T and NK cells expressing Th1/Tc1/Type 1 cytokines in culture-positive lymph node tuberculosis

PONE-D-21-36622R2

Dear Dr. Kathamuthu,

We’re pleased to inform you that your manuscript has been judged scientifically suitable for publication and will be formally accepted for publication once it meets all outstanding technical requirements. Please forgive the slow response as there were some editorial issues that needed to be resolved.

Kind regards,

Angelo A. Izzo

Academic Editor

PLOS ONE
---

## [Editor Report · Acceptance letter]

18 May 2022

PONE-D-21-36622R2 

Dominant expansion of CD4^+^, CD8^+^ T and NK cells expressing Th1/Tc1/Type 1 cytokines in culture-positive lymph node tuberculosis 

Dear Dr. Kathamuthu:

I'm pleased to inform you that your manuscript has been deemed suitable for publication in PLOS ONE. Congratulations! Your manuscript is now with our production department. 

Kind regards, 

on behalf of

Dr. Angelo A. Izzo 

Academic Editor

PLOS ONE